# Cocaine Regulates NLRP3 Inflammasome Activity and CRF Signaling in a Region- and Sex-Dependent Manner in Rat Brain

**DOI:** 10.3390/biomedicines11071800

**Published:** 2023-06-23

**Authors:** Yan Cheng, Rachael Elizabeth Dempsey, Soheil Kazemi Roodsari, Dorela D. Shuboni-Mulligan, Olivier George, Larry D. Sanford, Ming-Lei Guo

**Affiliations:** 1Drug Addiction Laboratory, Department of Pathology and Anatomy, Eastern Virginia Medical School, Norfolk, VA 23507, USA; chengy@evms.edu (Y.C.); dempsere@evms.edu (R.E.D.); roodsask@evms.edu (S.K.R.); 2Sleep Laboratory, Department of Pathology and Anatomy, Eastern Virginia Medical School, Norfolk, VA 23507, USA; shubond@evms.edu (D.D.S.-M.); sanforld@evms.edu (L.D.S.); 3Center for Integrative Neuroscience and Inflammatory Diseases, Eastern Virginia Medical School, Norfolk, VA 23507, USA; 4Department of Psychiatry, School of Medicine, University of California San Diego, San Diego, CA 92093, USA; olgeorge@health.ucsd.edu

**Keywords:** self-administration, cocaine, neuroinflammation, cocaine use disorders, CRF signaling, NLRP3 inflammasome

## Abstract

Cocaine, one of the most abused drugs worldwide, is capable of activating microglia in vitro and in vivo. Several neuroimmune pathways have been suggested to play roles in cocaine-mediated microglial activation. Previous work showed that cocaine activates microglia in a region-specific manner in the brains of self-administered mice. To further characterize the effects of cocaine on microglia and neuroimmune signaling in vivo, we utilized the brains from both sexes of outbred rats with cocaine self-administration to explore the activation status of microglia, NOD-, LRR-, and pyrin domain-containing protein 3 (NLRP3) inflammasome activity, corticotropin-releasing factor (CRF) signaling, and NF-κB levels in the striatum and hippocampus (HP). Age-matched rats of the same sex (drug naïve) served as controls. Our results showed that cocaine increased neuroinflammation in the striatum and HP of both sexes with a relatively higher increases in male brains. In the striatum, cocaine upregulated NLRP3 inflammasome activity and CRF levels in males but not in females. In contrast, cocaine increased NLRP3 inflammasome activity in the HP of females but not in males, and no effects on CRF signaling were observed in this region of either sex. Interestingly, cocaine increased NF-κB levels in the striatum and HP with no sex difference. Taken together, our results provide evidence that cocaine can exert region- and sex-specific differences in neuroimmune signaling in the brain. Targeting neuroimmune signaling has been suggested as possible treatment for cocaine use disorders (CUDs). Our current results indicate that sex should be taken into consideration when determining the efficacy of these new therapeutic approaches.

## 1. Introduction

Substance use disorders (SUDs) remain a major public health concern around the world. For example, the psychostimulant cocaine is among the top abused drugs, and deaths from cocaine overdoses doubled between 2011 and 2016 [1]. Epidemiological studies show that SUDs, including cocaine use disorders (CUDs), are prevalent in both males and females [2], with sex-dependent progression and relapse patterns. During the last decade, the rate of increase in SUDs was significantly greater in females when compared to males [3], and females escalate their psychostimulant use faster than males [4]. Females are more likely to self-report intense highs and develop faster escalation in cocaine use [5,6]. Females also report more intense cravings and a higher likelihood of relapse with more severe mood abnormalities than males [6,7,8]. The neurobiological mechanisms underlying sex-differences in the disease course of CUDs remain unclear.

Accumulating evidence suggests that CUDs are neuroinflammation-related diseases and that neuroimmune dysregulation is inherent in their pathogenesis. Microglia, the resident brain macrophage, account for 10–15% of all brain cells and constitute the first defense against various immune insults. Increased microglial activation has been consistently found in CUDs [9,10]. Cocaine can activate microglia both in vitro and in vivo [11,12]. Cocaine and methamphetamine increase the expression of pro-inflammatory factors such as interleukin-1 beta (IL1β), interleukin 6 (IL6), tumor necrosis factor alpha (TNFα), and C-C motif chemokine ligand 2 (CCL2) in the prefrontal cortex (PFc) and nucleus accumbens (NAc) [13,14]. Enhanced IL1β levels in the ventral tegmental area (VTA) are involved in cocaine-mediated behavioral changes such as conditional place preference (CPP) and self-administration [15]. Minocycline, an inhibitor of microglial activation, mitigates/blocks reward-related behavioral changes induced by abused drugs [16,17]. Cocaine self-administering macaques showed a strong inflammatory response in the NAc [18]. In addition to animal work, the brains of addicts have revealed a close association between neuroinflammation and stimulant addiction, with a significant increase in activated microglia in chronic drug abusers [19,20]. Targeting neuroimmune signaling has been suggested as a potential treatment for preventing or ameliorating CUDs [21,22]. Whether neuroimmune dysregulation equally contributes to CUDs in males and females remains unexplored.

Several neuroimmune pathways have been shown to be involved in cocaine-mediated microglial activation. The NOD-, LRR-, and pyrin domain-containing protein 3 (NLRP3) inflammasome belongs to the superfamily of pattern recognition receptors [23], is abundantly expressed in microglia [24], and plays critical roles in microglia activation under various stimuli. NLRP3 activation needs two sequential signals: signal 1 for priming through increasing the mRNA and protein levels of NLRP3 as well as mRNA levels of IL1β; and signal 2 for assembling the NLRP3 inflammasome by promoting the binding of NLRP3, apoptosis-associated speck-like protein containing a CARD (ASC), and pro-caspase 1, which, in turn, cleaves pro-IL1β into mature (m) IL1β [25]. The toll-like receptor/nuclear factor kappa B (TLR/NF-κB) axis is well-known as signal 1, and ATP, reactive oxygen species (ROS), and Ca^2+^ may each directly serve as signal 2 [26,27]. Abnormal NLRP3 activity has been identified in multiple neuroinflammatory diseases including Alzheimer’s disease, stroke, and viral infections [28,29,30,31]. Additionally, the NLRP3 inflammasome can be upregulated by abused drugs including cocaine and is involved in cocaine-mediated microglia activation and behavioral changes [32,33]. Whether the regulation of NLRP3 by cocaine has sex-specific effects has not previously been investigated.

CRF is a neuronal hormone regulated by a variety of stressors [34,35] and has also been implicated in CUDs. CRF and its cognate receptors 1 and 2 (CRHR1/2) are expressed in both hypothalamus and extra-hypothalamic regions, such as the striatum, HP, cortex, and amygdala [36,37,38]. Extra-hypothalamic CRF, especially in the amygdala, is sensitive to cocaine regulation [39], whereas striatal CRF signaling has been extensively investigated with respect to potential cocaine-mediated reward effects [40]. However, potential sex-differences in cocaine-regulated CRF signaling in brain reward circuitry have been seldom explored.

To explore possible regional and sex differences in cocaine-mediated neuroimmune dysregulation, we examined neuroinflammation levels, NLRP3 inflammasome activity, NF-κB levels, and CRF signaling in the brains of male and female cocaine self-administered rats. Our results demonstrate that cocaine can induce microglial activation in the striatum and HP of both sexes. However, cocaine exerts region- and possible sex-specific effects on microglial activation markers, NLRP3 activity, and CRF signaling. Such differential effects induced by cocaine might underly the sex-dependent progression and relapse pattern of CUDs. Our results thus indicate that sex should be taken into consideration when developing therapeutic agents for CUDs that target neuroimmune molecules.

## 2. Materials and Methods

### 2.1. Rat Brain Tissues

Brains from male and female rats with a history of cocaine addiction-like behaviors and aged-matched naive controls were obtained from the cocaine biobank (https://www.cocainebiobank.org/home; accessed on 2 May 2021). Details on all surgical and behavioral procedures can be found in the original manuscript [41]. Briefly, the brains came from heterogeneous stock rats with a history of cocaine self-administration (n = 5) and age-matched controls. They received intrajugular catheterization and after recovery were allowed to self-administer cocaine (0.5 mg/kg/inj, FR1) for over 1 month, including 10 short access (2 h) sessions and 14 long access (6 h) sessions. Addiction-like behaviors were evaluated by measuring cocaine intake under a fixed ratio 1, the breaking point using a progressive ratio schedule of reinforcement, and continued responses despite a contingent mild foot shock. Animals for the current report were selected in such a way that they are representative of the original population of heterogeneous rats that are self-administering cocaine in the Biobank. In other words, the animals selected represent the spectrum of low, moderate, and severe addiction-like behaviors, with the majority of the subjects exhibiting moderate addiction-like behaviors. Animals were euthanized during the last session of self-administration after ~3 h of self-administration to capture the state of intoxication. The frozen brains were shipped on dry ice and immediately stored at −80 °C when received at EVMS. Each brain was cut equally into two halves (left and right hemispheres) and further dissected into the striatum and hippocampus (HP). Tissue from one side was used for protein extraction and the other side for total RNA extraction.

### 2.2. RNA Extraction, Reverse Transcription, and Quantitative Polymerase Chain Reaction (qPCR)

To extract RNA, approximately 100–200 microglia of brain tissue was directly added to 1 mL Trizol (Invitrogen, Waltham, MA, USA). Brain lysates were briefly sonicated (3–5 s) and incubated for 10 min on ice and then aspirated into new 1.5 mL microcentrifuge tubes with 0.2 mL of chloroform added. After vigorous vortexing, the samples were centrifuged at 10,000× *g* for 15 min at 4 °C. The upper aqueous phase was transferred to a new tube and 500 µL isopropyl alcohol was added. Samples were then incubated for 10 min and centrifuged again to precipitate total RNA. The total RNA was dissolved in DEPC-treated H_2_O and quantified. Reverse transcription reactions were performed using a Verso cDNA kit (Invitrogen). The reaction system (20 µL) included 4 µL 5× cDNA synthesis buffer, 2 µL dNTP mix, 1 µL RNA primer, 1 µL RT enhancer, 1 µL Verso enzyme Mix (Invitrogen), total RNA template 1.0 µg, and a variable volume of water. Reaction conditions were set at 42 °C for 45 min. qPCRs were performed by using SuperScript™ III Platinum™ One-Step qRT-PCR Kit (Invitrogen). Reaction systems were set up as follows: 10 µL Master mix, 1.0 µL primers and probes, and 1 µL cDNA and 8 µL distilled H_2_O. The 96-well plates were placed into a QS3 qPCR machine (Invitrogen) for program running. Rat primers for IL1β, IL6, CCL2, and TNFα were purchased from Invitrogen (Rn00580432, Rn01410330, Rn00580555, Rn99999017). Rat GADPH (Invitrogen, Rn01775763) served as internal control for quantification.

### 2.3. Western Blots

Brain tissues (100 microglia) were dissolved in RIPA buffers with proteinase and phosphatase inhibitors (Thermo Scientific, Waltham, MA, USA) and sonicated for 10 s on ice at 70% amplitude (Thermo Scientific). The brain homogenates were then incubated at 4 °C for 30 min, followed by 12,000 rpm centrifugation for 10 min. The supernatants were taken out and the protein concentrations were calculated through the BCA method. Equal amounts of the proteins (35 µg) were electrophoresed in a sodium dodecyl sulfate-polyacrylamide gel (160 V, 60 min) under reducing conditions followed by transfer to PVDF membranes (180 mA, 90 min). The blots were blocked with 3% nonfat dry milk in Tris-buffered saline (TBST). The Western blots were then incubated with indicated antibodies overnight at 4 °C. The next day, the membranes were washed and incubated with appropriate IRDye fluorescent mouse or rabbit second antibody for one hour at room temperature. After three washes with TBST, the membranes were put into the Odyssey^®^ Imaging System (LI-COR, Lincoln, NE, USA). for image development and the intensity of the fluorescent band was quantified using Image Studio™ Software (version 5.2). After imaging, the membranes were re-probed by β-actin for normalization. The following antibodies were used at the indicated concentration in our studies: microglial activation marker CD11b (1:2000, NBP2-19019), astrocyte activation marker GFAP (1:5000; Abcam (Cambridge, United Kingdom), ab7260); NLRP3 (1:2000, AdipoGen (San Diego, CA, USA), AG-20B-0014-C100); Caspase 1 (1:1000, Proteintech (Rosemont, IL, USA), 22915-1-AP); IL1β (1:1000, Proteintech, 26048-1-AP); CRF (1:1000, Proteintech, 10944-1-AP); CRFR1 (1:1000; Sigma (St. Louis, MO, USA), SAB4500465); CRFR2 (1:1000, Sigma, SAB4500467); NF-κB (1:2000, Proteintech, 10745-1-AP); ASC (1:1000, Santa Cruz Biotech (Dallas, TX, USA), sc-514414); β-actin (Santa Cruz; 1:2000, sc-8432 or Sigma; 1:10,000, A2066). Secondary antibodies were purchased from Li-COR company, including IRDye^®^ 680RD Donkey anti-Mouse (1:5000) or rabbit IgG and IRDye^®^ 800CW Donkey anti-Mouse or rabbit IgG (1:5000).

### 2.4. Statistical Analysis

All data are expressed as means ± the standard error of the mean (SEM). Data were statistically evaluated by unpaired two-tailed student *t*-tests using GraphPad Prism 9 (La Jolla, CA, USA). Each group includes at least four brain samples. Tests with probability levels of <0.05 were considered statistically significant.

## 3. Results

### 3.1. Cocaine Upregulates Neuroinflammation in the Striatum and HP of Male and Female Rats

We monitored neuroinflammation levels in the striatum and HP dissected from the brains of male and female self-administrated rats. In male striatum, cocaine significantly increased the levels of IL1β (15.88 ± 3.88 fold, * *p* < 0.05), TNFα (2.10 ± 0.47 fold, * *p* < 0.05), and CCL2 (2.50 ± 0.67 fold, * *p* < 0.05) but not IL6 (1.26 ± 0.27 fold, *p* > 0.05) (Figure 1A). In female striatum, cocaine increased the levels of IL1β (10.38 ± 1.76 fold, * *p* < 0.05) while the other three pro-inflammatory mediators did not show significant changes: IL6 (1.19 ± 0.41 fold, *p* > 0.05), TNFα (1.47 ± 0.90 fold, *p* > 0.05), and CCL2 (1.20 ± 0.61 fold, *p* > 0.05) (Figure 1B). These results suggested that cocaine upregulated striatal neuroinflammation levels more strongly in males than in females. In male HP, cocaine increased the levels of IL1β (12.55 ± 5.01 fold, * *p* < 0.05), TNFα (3.54 ± 0.92 fold, * *p* < 0.05), and CCL2 (2.21 ± 0.41 fold, * *p* < 0.05) but not for IL6 (1.08 ± 0.24 fold, *p* > 0.05) (Figure 1C). In females, cocaine increased the levels of IL1β (9.7 ± 0.98 fold, * *p* < 0.05), IL6 (1.57 ± 0.19, * *p* < 0.05); TNFα (1.80 ± 0.27 fold, * *p* < 0.05), and CCL2 (2.32 ± 0.58 fold, * *p* < 0.05) (Figure 1D). Similar to the striatum, cocaine increased neuroinflammation levels in the HP with higher fold expression in males than in females. We also compared neuroinflammation levels in the striatum and HP in control rats. Interestingly, striatal pro-inflammatory mediators showed significant higher levels in males than in females: IL1β (2.08 ± 0.38 fold, * *p* < 0.05); IL6 (2.74 ± 0.37 fold, * *p* < 0.05); TNFα (2.05 ± 0.39 fold, * *p* < 0.05) but not for CCL2 (1.47 ± 0.39 fold, *p* > 0.05) (Figure 1E). However, in the HP, there were no differences in the levels of these four pro-inflammatory mediators between males and females (*p* > 0.05) (Figure 1F). Thus, the higher increase on striatal IL1β in male rats might come from its higher basal levels. Taken together, these findings indicated that cocaine increases neuroinflammation levels in a region- and sex-dependent (at least in the hippocampus) manner in rat brain.

### 3.2. Cocaine Increases Microglial Markers in the Male Striatum

Increased neuroinflammation levels imply glial activation. Previous studies showed that cocaine could activate both microglia and astrocytes in vitro; therefore, we assessed brain levels of the microglia activation marker, CD11b, and the astrocytes marker, GFAP. In the striatum, cocaine significantly increased CD11b levels in male (1.98 ± 0.23 fold, * *p* < 0.05) but not in female rats (1.06 ± 0.06 fold, *p* > 0.05) (Figure 2A). Cocaine had no obvious effects on GFAP levels in either males (1.05 ± 0.02 fold, *p* > 0.05) or females (1.05 ± 0.04 fold, *p* > 0.05) (Figure 2B). In the HP, we did not observe significant changes on CD11b or GFAP levels in either sex (*p* > 0.05) (Figure 2C,D). These results showed that cocaine could increase CD11b levels in a region- and sex-dependent manner but produced no significant effects on GFAP levels in the regions we examined.

### 3.3. Cocaine Differentially Upregulates NLPR3 Inflammasome Activity in Rat Brain

NLRP3 inflammasome is involved in cocaine-mediated behavioral changes [33]. Here, we assessed the activity of the NLPR3 inflammasome by determining the protein levels of NLRP3, pro- and mature caspase 1, and pro- and mature IL1β in cocaine-exposed rat brains of both sexes. NLRP3 was significantly increased in male but not female striatum (1.42 ± 0.09 fold, * *p* < 0.05; 0.99 ± 0.07 fold, ns) (Figure 3A). Similarly, cocaine significantly increased the levels of mature caspase 1 and mature IL1β in male striatum (1.76 ± 0.11 fold, * *p* < 0.05; 2.24 ± 0.25 fold, * *p* < 0.05, respectively) (Figure 3B,C). We did not observe upregulation of NLRP3, mature caspase 1, and mature IL1β in the striatum of females (*p* > 0.05) (Figure 3B,C). In male HP, we observed significant upregulation of NLRP3 levels (1.96 ± 0.08 fold * *p* < 0.05) (Figure 4A) but there were no significant changes in the levels of mCasp1 and mIL1β, indicating that the NLPR3 inflammasome was primed but not fully activated. In contrast, we did not find significant changes in female NLRP3, but there were significant upregulations in mCasp1 (1.84 ± 0.19 fold, * *p* < 0.05) (Figure 4B) and mIL1β (1.56 ± 0.08 fold, * *p* < 0.05) (Figure 4C). ASC dimer and oligomerization is critical for NLRP3 inflammasome activation. We thus monitored ASC level in the striatum and HP of male and female rats with cocaine self-administration. Cocaine significantly increased ASC dimer levels in the striatum of male and female striatum (1.84 ± 0.14 fold, * *p* < 0.05 and 1.57 ± 0.09 fold, * *p* < 0.05, respectively) (Figure 5A,B). However, we did not observe significant changes on ASC monomer, dimer, and oligomer in the HP of male and female rats with cocaine exposure (Figure 5C,D). These results indicate that other inflammasomes might be involved in IL1β cleavage and mIL1β production. Taken together, our results indicated that cocaine primes and activates the NLRP3 inflammasome in a region- and sex-dependent manner.

### 3.4. The Effects of Cocaine on CRF Signaling and NF-κB Levels in Rat Brain

CRF and its cognate receptors, CRFR1 and CRFR2, are expressed in the striatum and HP. CRF signaling has been shown to promote cocaine-mediated reward effects [40] and CRF has the ability to activate microglia in vitro [42]. Here, we determined the effects of cocaine on CRF signaling in vivo. In the striatum, cocaine significantly increased CRF levels in males (1.42 ± 0.07 fold, * *p* < 0.05) but not in females (0.96 ± 0.05 fold, *p* > 0.05) (Figure 6A). We did not detect significant changes in CRFR1 and CRFR2 in the striatum of either sex (Figure 6B,C) (*p* > 0.05). In the HP, there were no significant changes in levels of CRF, CRFR1, CRFR2 in either sex (Figure 7A–C) (*p* > 0.05). Taken together, our results revealed that CRF signaling was probably upregulated only in the striatum of cocaine-treated males.

NF-κB is a master regulator for the transcription of pro-inflammatory mediates and has been implicated in cocaine-mediated microglia activation. NF-κB-mediated signaling could serve as signal 1 for NLRP3 priming. We thus determined the levels of NF-κB in the striatum and HP of both sexes. Cocaine increased striatal NF-κB levels in both male and females (1.52 ± 0.11 fold, * *p* < 0.05 and 1.79 ± 0.14 fold, * *p* < 0.05, respectively) (Figure 8A). In the HP, cocaine also increased NF-κB levels in males (1.42 ± 0.05 fold, * *p* < 0.05) and females (1.68 ± 0.12 fold, * *p* < 0.05) (Figure 8B). Our findings suggest that cocaine upregulated NF-κB signaling in a ubiquitous manner.

## 4. Discussion

Epidemiological studies show that CUDs exhibit sex-dependent differences in progression and relapse; however, the mechanisms driving these differences are unknown. Accumulating evidence suggests that CUDs are neuroinflammation-related brain diseases and that targeting neuroimmune signaling may provide an alternative therapeutic approach for treating SUDs [21,22]. Cocaine-mediated immune responses may be sexually dimorphic [43]. To further characterize the effects of cocaine on microglia and neuroimmune signaling, we used a well-established rodent model of outbred rats (both sexes) with cocaine self-administration to investigate the activation status of microglia and three neuroimmune pathways that have been shown to be regulated by cocaine: NLRP3 inflammasome, CRF signaling, and NF-κB signaling. Our results showed that cocaine increased neuroinflammation levels in the striatum and HP in both sexes with evidence of generally greater levels in males. Thus, cocaine exerts differential sex- and region-specific effects on different neuroimmune signaling. Our results provide additional evidence supporting roles for neuroinflammation and neuroimmune signaling in the pathogenesis of SUDs of both sexes and also underlines the importance of including both sexes in future studies.

In experimental animal models, cocaine can be administered passively (via I.P. injections) or actively through self-administration. In self-administration, rodents are given the opportunity to “choose” to take the drugs of interest by engaging in a learned behavior (lever pressing or nose-poking) rather than passively receiving experimenter-administered drug. This rodent model thus better recapitulates human addictive behavior that occurs in the natural environment and remains the “gold standard” for behavioral experiments in the field of CUDs [44,45]. It has been shown that different administration routes could strikingly alter the effects of cocaine on gene expression [46]. Of note, the brains utilized here were from outbred rats, meaning their genetic background is heterogeneous, which is more analogous to humans than inbred strains. Therefore, our results may be more translationally significant with respect to genetic variability and provide additional evidence that neuroinflammation is involved in the pathogenesis of CUDs.

The activation of microglia in vivo in our study showed varying results depending on the different approaches employed. Through qRT-PCR, we monitored neuroinflammation upregulation (i.e., microglia activation) in the striatum and HP of both sexes. However, we observed microglia activation only in male striatum when we monitored the levels of CD11b, a commonly used microglia marker [47]. Such discrepancy may be derived from the different sensitivity of these two approaches. The QRT-PCR approach is to detect the mRNA levels of cytokines and chemokines, which are very sensitive to mRNA levels, while CD11b was monitored by western blot for its protein expression. The highly basal levels of CD11b might be the factor masking the changes on CD11b. Increased neuroinflammation can also be attributed to astrocyte activation. We checked GFAP levels but did not observe significant differences between any groups. Thus, our results favor the interpretation that microglia were activated and contributed to the increased neuroinflammation in cocaine self-administered rats. Previous results showed that there was actually a reduction in astrocyte GFAP levels in the striatum of rats with cocaine self-administration and extinction [48], which also suggests that astrocytes do not contribute to the increased neuroinflammation produced by cocaine self-administration. In future studies, we could purify adult microglia and astrocytes and directly monitor their inflammatory status to further test this hypothesis.

Brain-resident macrophages determine the intensity of immune responses and inflammation levels in the brain and can be influenced by various stimuli. Microglia characteristics differ across the brain and vary in the density, morphology, and functional status depending on region [49,50]. Single-cell (sc) RNA sequencing has also revealed that microglia are heterogeneous in both physiological and pathological conditions based on their gene expression profiles [51]. Microglia also are sexually dimorphic throughout the life span under unstimulated conditions. In the early postnatal stage, gene ontology analysis at postnatal (P) day 20 revealed that inflammatory response genes were upregulated in female microglia compared with males indicating that microglia are in a more primed state in females at this age [52]. In adults (3-month-old), RNA sequencing suggests that more inflammatory-related genes are expressed in male brains than in females [53,54]. Furthermore, immune responses of microglia are sex dependent. LPS could trigger a greater release of inflammatory cytokines from microglia isolated in the male rat brain (P0/P1) than from females [55]. In adulthood, LPS induces more microglia activation in males than in females [56]. Similarly, LPS increased Iba1+ cell number and area in 3-month-old male, but not female mice [57]. In contrast, LPS increased the levels of inflammatory cytokines to a greater extent in aged female compared with male mice [58]. Overall, these findings show that microglia and their immune responses are age- and sex-dependent and suggest that sex is a biological factor regulating the status of microglia and immune responses.

Although there is ample preclinical and clinical evidence showing a sex-dependent pattern on SUDs, the mechanisms underlying this phenomenon remain much elusive. Microglia are sexually dimorphic in response to inflammatory stimulus; thus, it is possible that abused drugs can also induce differential immune responses of microglia which underlies the sex dimorphic pathogenesis of SUDs. Indeed, several abused drugs have been shown to induce microglial activation in a sex-dependent manner. Alcohol has been well-known to sex-dependently induce brain transcriptomic changes related to alcohol use disorder [59]. For example, binge ethanol drinking can induce greater changes to the inflammatory cytokines profile in adolescent females than in adolescent males [60]. However, another investigation showed binge-level alcohol drinking inhibited TNF receptor 2 signaling in adult female mice but activated this pathway in adult male mice [61]. The discrepancy between these two studies may be due to mouse age (adolescence vs. adult) or alcohol regimen they selected. Opioid-like drugs have the ability of inducing sexually dimorphic neuroimmune responses in mice. Chronic oxycodone and withdrawal-treated male mice had higher protein levels of pro-inflammatory cytokines/chemokines and growth factors including IL1β, IL2, and IL7 in the PFC as compared to their female counterparts. In contrast, reduced levels of pro-inflammatory cytokines/chemokines IL1β, IL6, and CCL11 were observed in the NAc of oxycodone and withdrawal-treated males as compared to female mice [62]. One caveat of previous studies is that they did not compare the basal levels of pro-inflammatory cytokine between sexes. Thus, the sex-dependent difference on abused-drug-mediated inflammatory responses might be partly induced by the difference in their basal levels. Direct comparisons on the effects of psychostimulants such as cocaine on neuroimmune signaling between sexes remain scarce. However, a relevant study showed that higher neuroimmune response can be positively related to cocaine-mediated reward effects between sexes in the context of traumatic brain injury (TBI) model. Adolescent TBI increased susceptibility to the rewarding effects of cocaine in male adult mice in parallel with augmented inflammatory profiles, increased microglial phagocytosis of neuronal proteins, and decreased neuronal spine density in the NAc. On the contrary, female adult mice with higher levels of female sex hormones in the time of TBI showed reduced sensitivity to the rewarding effects of cocaine, with significantly reduced microglial activation and phagocytosis of neuronal proteins within the NAc [63]. According to the statistics of drug consumption by sex, men are more prone to psychostimulant use than women (United Nation Office on Drugs and Crime. World Drug Report 2018). However, once women begin to consume drugs, they tend to progress faster from use to abuse, a phenomenon known as telescoping [64]. It will be interesting to directly compare the neuroimmune responses between sexes in the different stages of cocaine use.

Among the four pro-inflammatory mediators we examined, IL1β was the most sensitive to cocaine exposure, supporting its critical roles in cocaine-mediated reward effects in a manner consistent with previous findings [15]. The mechanisms underlying cocaine-mediated sex differences in neuroinflammation upregulation are unclear. Based on previous studies [53,54], it is possible that the pro-immune state of microglia in male striatum could make microglia more sensitive in response to an incoming stimulus. Therefore, we directly compared neuroinflammation levels in the striatum and HP between sexes. The results showed that higher neuroinflammation levels can be observed in male striatum than in females, which is consistent with our hypothesis. However, we did not see sex differences in neuroinflammation levels in HP. Sex hormones have been extensively investigated for regulating microglia activation in various conditions. For example, progesterone attenuates brain inflammatory responses in rodent models [65]. Gonadal hormones differentially regulate sex-specific stress effects on glia in the medial prefrontal cortex [66]. It is thus possible that sex hormones are responsible for the difference in cocaine-mediated immune responses in the HP. This hypothesis needs further investigation.

A novelty for our finding is that cocaine could exert differential region- and sex-dependent patterns for the three neuroimmune signaling pathways we examined: NLRP3 inflammasome activation in male striatum and female HP; CRF signaling activation in male striatum; and NF-κB activation in both regions for both sexes. The reasons underlying the differential effects on NLRP3 and CRF signaling remain unknown and should be further explored in future studies. Interestingly, we observed NF-κB activation in both regions of both sexes; however, the neuroinflammation levels were sex- and region- dependent. This phenomenon implies that multiple factors and signaling coordinately decided the final results of neuroimmune response, which may include basal microglia activation status, age, sex, and local microenvironment. Our results also indicate that microglia activation across brain regions could have different sex-dependent neuroimmune mechanisms. The striatum and HP play critical roles in various stages of CUDs: the striatum is more involved in the initiation stage of CUDs and HP is more relevant to the withdrawal and relapse stages [67]. For example, upregulation of the NLRP3 inflammasome in male striatum and in female HP induced by cocaine indicate that NLRP3 could have sex-dependent roles in contributing to the pathogenesis of CUDs. Such results indicate that the addiction stage of CUDs and sex should be considered in assessing the potential of neuroimmune targeting for CUDs treatment.

## 5. Conclusions

Cocaine increases neuroinflammation in both sexes of rats, and cocaine regulates neuroimmune signaling in both a region- and a sex-specific manner. Sex should be considered as a relevant factor in testing therapies for CUDs.

## Figures and Tables

**Figure 1 biomedicines-11-01800-f001:**
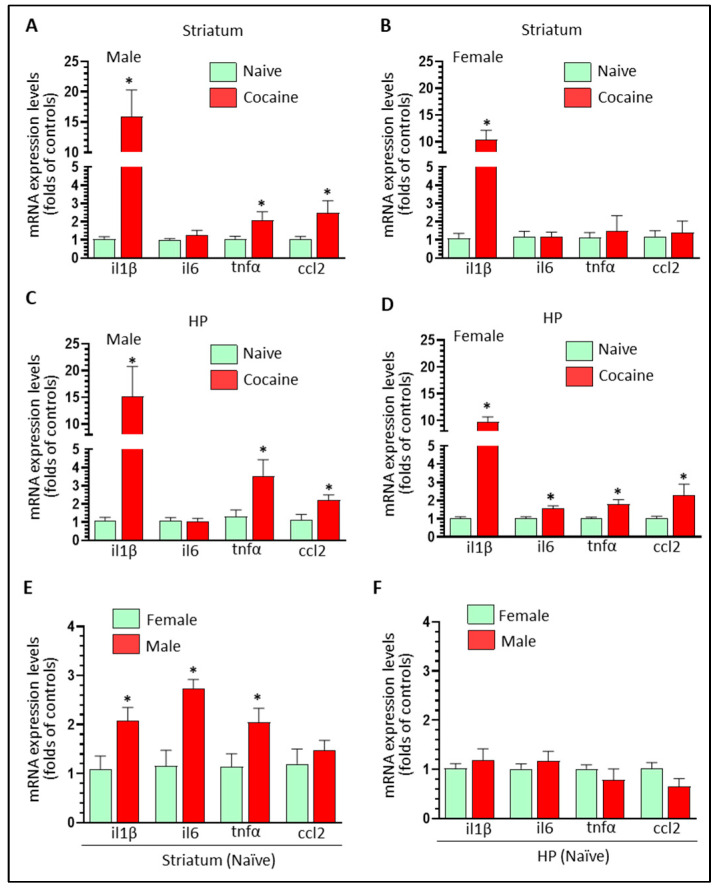
Cocaine self-administration upregulates neuroinflammation in both striatum and HP in male and female rats. (**A**) Cocaine signficantly increased neuroinflammation in male striatum (n = 5, * *p* < 0.05 vs. controls). (**B**) Cocaine signficantly increased neuroinflammation in female striatum (n = 5, * *p* < 0.05 vs. controls). (**C**) Cocaine signficantly increased neuroinflammation in male HP (n = 5, * *p* < 0.05 vs. controls). (**D**) Cocaine signficantly increased neuroinflammation in female HP (n = 5, * *p* < 0.05 vs. controls). (**E**) In basal conditions, neuroinflammation levels in male striatum was signficantly higher than in female striatum (n = 5, * *p* < 0.05). (**F**) In basal conditions, there was no signfificant sex differences in neuroinflammatoin in the HP (n = 5).

**Figure 2 biomedicines-11-01800-f002:**
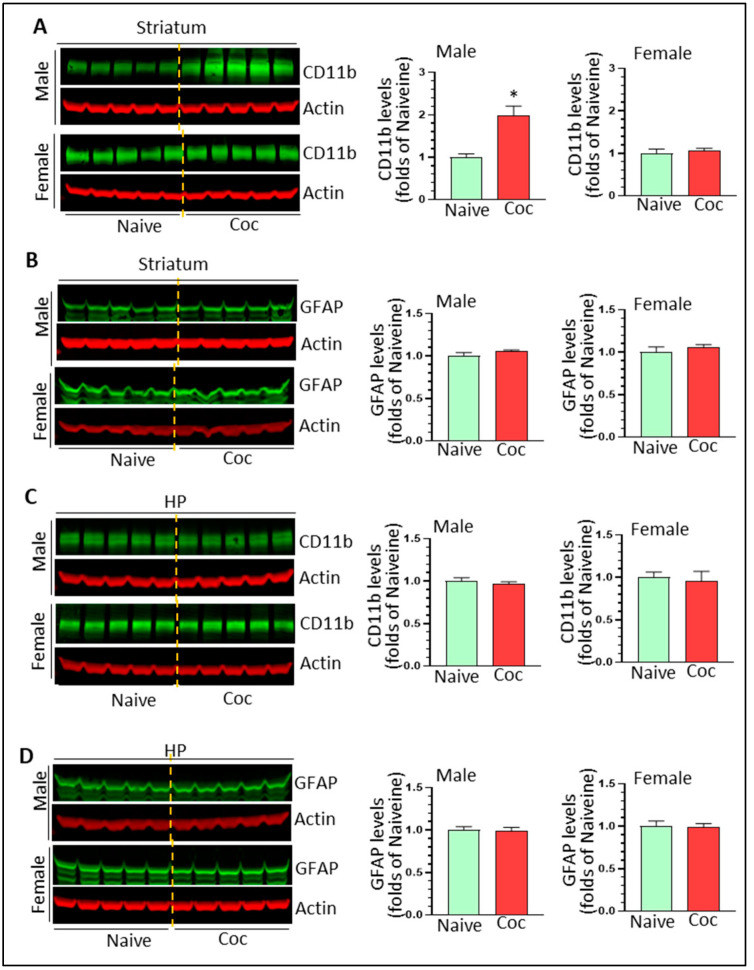
Cocaine self-administration upregulates microglial activation markers in male striatum. (**A**) Cocaine increases CD11b levels in male but not in female striatum (n = 5, * *p* < 0.05 vs. controls). (**B**) Cocaine has no significant effects on GFAP levels in striatum of either sex (n = 5). (**C**) Cocaine has no significant effects on CD11b levels in HP of either sex (n = 5). (**D**) Cocaine has no significant effects on GFAP levels in HP of either sex (n = 5).

**Figure 3 biomedicines-11-01800-f003:**
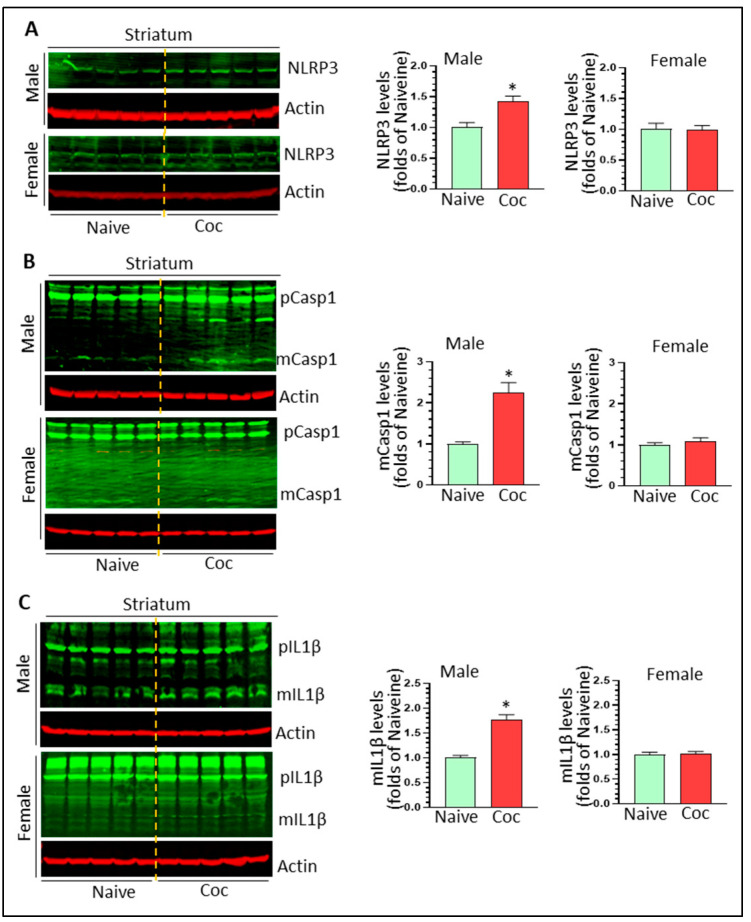
Cocaine increases the NLRP3 inflammasome in male striatum. (**A**) Cocaine significantly increases NLPR3 levels in male but not female striatum (n = 5, * *p* < 0.05). (**B**) Cocaine significantly increases mCasp 1 levels in male but not female striatum (n = 5, * *p* < 0.05). (**C**) Cocaine significantly increases mIL1β levels in male but not female striatum (n = 5, * *p* < 0.05).

**Figure 4 biomedicines-11-01800-f004:**
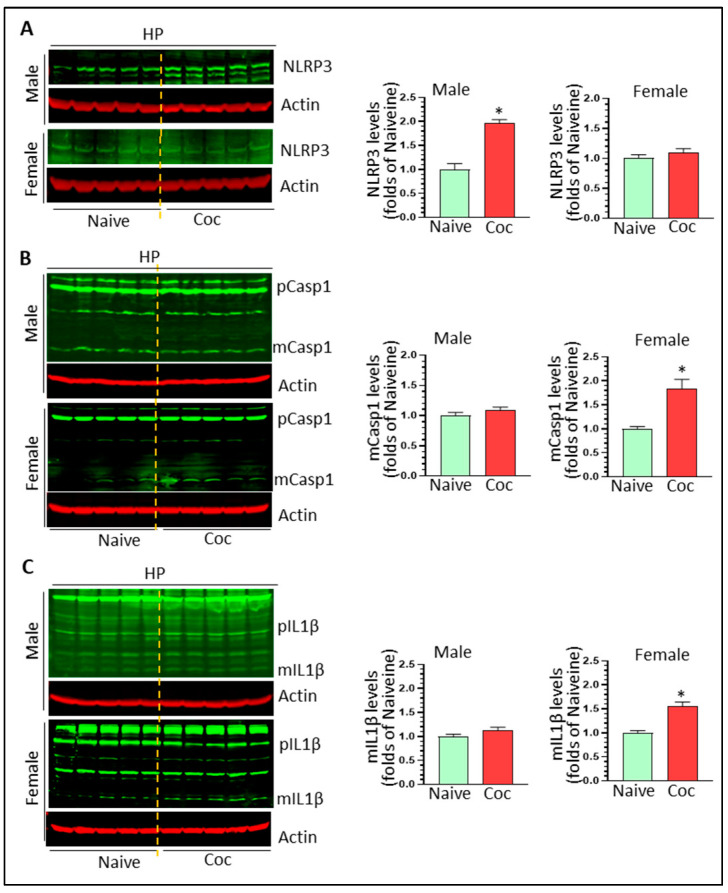
Cocaine increases mCasp1 and mIL1β levels in female HP. (**A**) Cocaine significantly increases NLPR3 levels in male striatum but not female striatum (n = 5, * *p* < 0.05). (**B**) Cocaine significantly increases mCasp 1 levels in female but not male HP (n = 5, * *p* < 0.05). (**C**) Cocaine significantly increases mIL1β levels in female but not male HP (n = 5, * *p* < 0.05).

**Figure 5 biomedicines-11-01800-f005:**
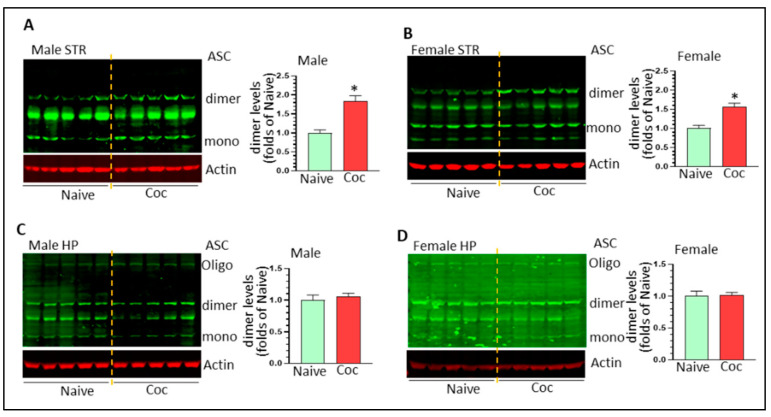
Cocaine increases ASC dimer in the striatum of male and female rats. (**A**) Cocaine significantly increases ASC dimer levels in male striatum (n = 5, * *p* < 0.05); (**B**) Cocaine significantly increases ASC dimer levels in female striatum (n = 5, * *p* < 0.05); (**C**) Cocaine has no significant effects on ASC levels in male HP; (**D**) Cocaine has no significant effects on ASC levels in female HP.

**Figure 6 biomedicines-11-01800-f006:**
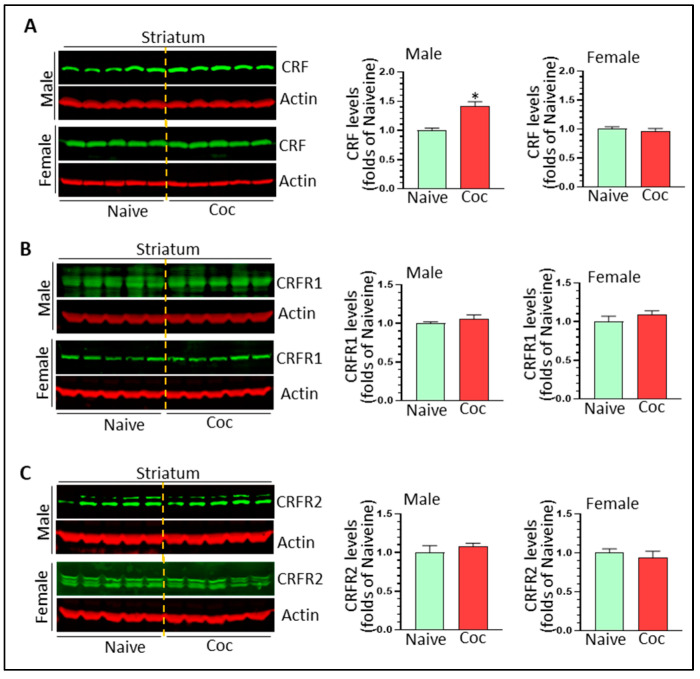
The effects of cocaine on CRF signaling in the striatum of both sexes. (**A**) Cocaine significantly increased CRF levels in male but not female striatum (n = 5, * *p* < 0.05). (**B**) Cocaine had no effects on CRFR1 in the striatum of either sex. (**C**) Cocaine had no effects on CRFR2 in the striatum of either sex.

**Figure 7 biomedicines-11-01800-f007:**
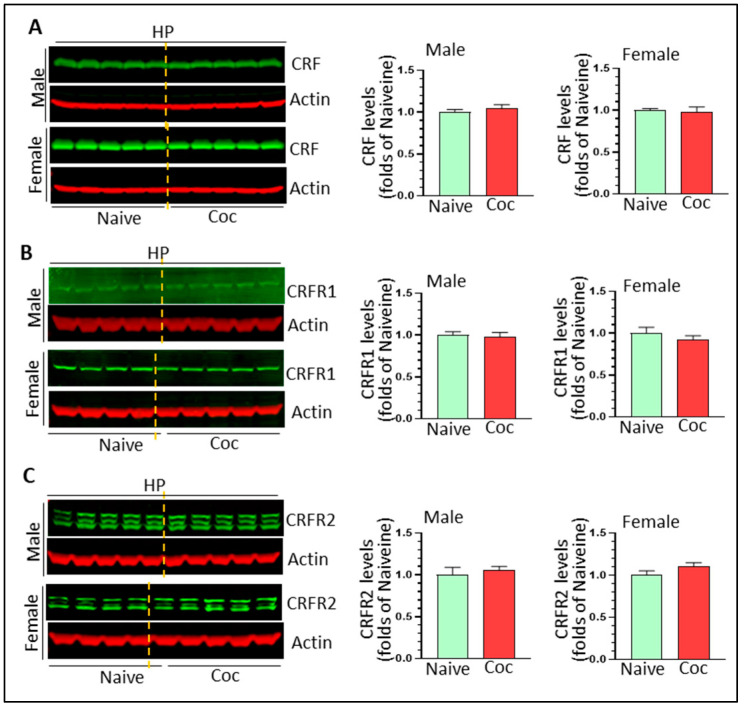
The effects of cocaine on CRF signaling in the HP of both sexes. (**A**) Cocaine has no effects on CRF levels in the HP of either sex. (**B**) Cocaine has no effects on CRFR1 in the HP of either sex. (**C**) Cocaine had no effects on CRFR2 in the striatum of either sex.

**Figure 8 biomedicines-11-01800-f008:**
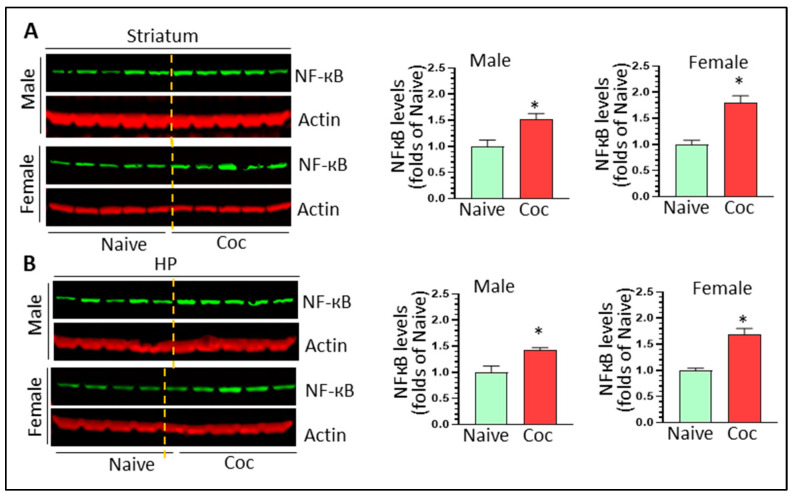
The effects of cocaine on NF-κB levels in the brains of both sexes. (**A**) Cocaine significantly increases NF-κb levels in the striatum of both sexes (n = 5, * *p* < 0.05). (**B**) Cocaine significantly increases NF-κb levels in the HP of both sexes (n = 4–5, * *p* < 0.05).

## Data Availability

Not applicable.

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
