# Peer review of "Cocaine Regulates NLRP3 Inflammasome Activity and CRF Signaling in a Region- and Sex-Dependent Manner in Rat Brain"

_biomedicines, 2023, doi:10.3390/biomedicines11071800_

Round 1

Reviewer 1 Report

In this study, Cheng and colleagues used the brains of outbred rats self-administered cocaine vs. saline from the bank and analyzed them for markers of neuroinflammation. To this end, they split each brain into two hemispheres and isolated the striatum and hippocampus from each half, and used one for WB analysis and the other half for rtPCR. They found that animals self-administering cocaine had higher levels of neuroinflammatory markers in the two brain regions, and in some cases, these changes were sex-specific. Overall, the manuscript is well-written and the study is important; however, given the nature of the data, which is rather correlative than causal. Also, the changes occurred in some markers, but not others. Thus, the authors should not make strong statements such as "strong evidence, critical roles, etc." The authors should attempt to block one of these pathways to see if that will prevent self-administration or not; otherwise, it is unclear whether the changes observed are due to cocaine self-administration or other actions of cocaine, given cocaine can increase the level of monoamines. Their metabolites can cause oxidative stress and neuroinflammation. Other specific comments are listed below:

Major:

1. Did all rats exhibit comparable self-administration? Did the control group self-administer saline to the same extent as the cocaine group? If not, if the authors control for the difference in operant responding? Is the response due to cocaine or cocaine and operant exposure?

2. If male and female are self-administering cocaine to the same extent, why there are differences in some of these markers. Are these changes related to cocaine self-administration, or there were differences in male and female, as shown in Figure 1E .

3. If there is already a sex-related difference and no difference in the magnitude of the iL1beta mRNA expression in response to cocaine between males and females, can you claim that "cocaine increases neuroinflammation levels in a region- and sex-dependent manner in rat brain."?

4. If NF-кB is upregulated, but there are not many changes in inflammatory markers in females, what does that mean, especially if these female rats self-administering cocaine (see also item 1, above)?

5. Line 295, the authors discuss the discrepancy in this sentence. Please elaborate on this, what the discrepant results have to do with the sensitivity of the two approaches? Are the two approaches measuring different chemicals, etc.

Minor:

1. Please do not abbreviate microglia; it can be confused with magnesium.

2. All other abbreviations need to be spelled out when they appear the first time in the text.

3. Line 151, please change to "Each group includes".

4. Section 3.1., only IL1beta changed in the females; can you still state that cocaine upregulates neuroinflammation?

5. Line 265, please change diving to driving.

The manuscript is well written; a few typos were detected, which can be corrected in the revised version or by the editor.

Author Response

In this study, Cheng and colleagues used the brains of outbred rats self-administered cocaine vs. saline from the bank and analyzed them for markers of neuroinflammation. To this end, they split each brain into two hemispheres and isolated the striatum and hippocampus from each half, and used one for WB analysis and the other half for rtPCR. They found that animals self-administering cocaine had higher levels of neuroinflammatory markers in the two brain regions, and in some cases, these changes were sex-specific. Overall, the manuscript is well-written and the study is important; however, given the nature of the data, which is rather correlative than causal. Also, the changes occurred in some markers, but not others. Thus, the authors should not make strong statements such as "strong evidence, critical roles, etc." The authors should attempt to block one of these pathways to see if that will prevent self-administration or not; otherwise, it is unclear whether the changes observed are due to cocaine self-administration or other actions of cocaine, given cocaine can increase the level of monoamines. Their metabolites can cause oxidative stress and neuroinflammation. Other specific comments are listed below:

Response:  Agree.  We have tuned our voice down.

Major:

  1. Did all rats exhibit comparable self-administration? Did the control group self-administer saline to the same extent as the cocaine group? If not, if the authors control for the difference in operant responding? Is the response due to cocaine or cocaine and operant exposure?

Response: We requested brain samples from the cocaine biobank for this project. The whole schematic for rodent selection, surgery, SA regimen, and control groups for cocaine biobank have been well-reviewed in previous publication (Carrette, et al, 2021). The brief summary of the approach has been added in the method. The control rats were drug naïve, age- and sex-matched ones without catheter implantation and saline self-administration (We made a mistake in previous description and now this has been corrected). We understand the control group is not the best for biochemical analysis since we can not exclude the potential effects of surgery and operant operation on neuroimmune signaling in this descriptive project. However, previous publication showed that there was no significant difference on c-fos expression between drug native (without surgery and operation) and saline self-administration group (Buffalari DM, et al., Neuroscience, 2016).  Also, a rat group with passively saline injection were used as control for cocaine self-administration to investigate the effects of cocaine on anxiety behaviors (White SL, et al., 2015). Thus, different types of control for drug self-administration are acceptable for descriptive/initial studies. Based on these literatures and our own experience on saline self-administration on mice (Burkovetskaya ME, 2020, Neurosci. Lett), we believe that the results on neuroimmune signaling should be mainly derived from cocaine action but not from surgery and operant exposure.                     

  1. If male and female are self-administering cocaine to the same extent, why there are differences in some of these markers. Are these changes related to cocaine self-administration, or there were differences in male and female, as shown in Figure 1E.

Response: Abused drugs have been known to induce sex-dimorphically neuroimmune responses, especially for alcohol.  The underlying mechanisms remain much elusive.  Based on the records, females on average self-administered more cocaine than males in this paradigm, the addiction index used to select animals was normalized per sex in order to have the same proportion of male and females in each request from the biobank.  Therefore, we believe such sex-dependent neuroimmune signaling are mainly due to the sex but not from the cocaine self-administration.        

 3. If there is already a sex-related difference and no difference in the magnitude of the iL1beta mRNA expression in response to cocaine between males and females, can you claim that "cocaine increases neuroinflammation levels in a region- and sex-dependent manner in rat brain."?

Response: Yes, there is already difference on il1beta mRNA in the striatum between male and female and cocaine increased the similar magnitude on il1beta mRNA levels between sexes.  However, in addition to il1beta mRNA, other cytokines and chemokines also contribute to neuroinflammation levels.  TNFalpha and CCl2 were increased by cocaine in the striatum of male but not in female.  Also, the protein levels of mature IL1beta were increased by cocaine exposure in male striatum but not in female.  Therefore, we argue that the statement “cocaine increases neuroinflammation levels in a region- and sex-dependent manner in rat brain” is still applicable.

  1. If NF-кB is upregulated, but there are not many changes in inflammatory markers in females, what does that mean, especially if these female rats self-administering cocaine (see also item 1, above)?

Response: Yes, we observed ubiquitous NF-кB upregulation in all male and female rats with cocaine self-administration.  NF-кB is a well-known transcription factor for upregulating neuroimmune signaling. However, the mechanisms underlying no much changes on inflammatory markers in female remain much unknown.  It is possible that neuroinflammation is coordinately regulated by multiple pro-, and anti- inflammatory pathways and some anti-inflammatory pathways have been also activated in female rats with cocaine exposure. Such possibility needs further investigations. Another possibility is that sex hormones modulate neuroinflammation levels.  More discussions addressing this issue have been added in the main context.    

  1. Line 295, the authors discuss the discrepancy in this sentence. Please elaborate on this, what the discrepant results have to do with the sensitivity of the two approaches? Are the two approaches measuring different chemicals, etc.

Response:  Yes, more discussion has been added in the main context. 

 Minor:

  1. Please do not abbreviate microglia; it can be confused with magnesium.

Response: Corrected!

  1. All other abbreviations need to be spelled out when they appear the first time in the text.

Response: Yes.

  1. Line 151, please change to "Each group includes".

Response: Corrected.

  1. Section 3.1., only IL1beta changed in the females; can you still state that cocaine upregulates neuroinflammation?

Response: Yes, only IL1beta mRNA upregulation was observed in female striatum. However, all four cytokines and chemokines were increased in female hippocampus.    

  1. Line 265, please change diving to driving.

Response: Sorry for the overlook.  Corrected. 

Reviewer 2 Report

In the present studies, the authors examined the effects of NOD-, LRR- and pyrin domain-containing protein 3 (NLRP3) inflammasome activity and corticotropin-releasing factor (CRF) signaling in self-administrative rats in response to cocaine. They found that cocaine activates microglia (Mg) in a region-specific manner in the brains of self-administered rats. To further characterize the effects of cocaine on Mg and neuroimmune signaling in vivo, they utilized the brains from both sexes of outbred rats with cocaine self-administration to explore the activation status of Mg, NLRP3 inflammasome activity, CRF signaling, and NF-кB levels in the striatum and hippocampus (HP). Saline self-administered rats of the same sex served as controls. They further found that cocaine increases neuroinflammation in the striatum and HP of both sexes with relatively higher fold increases in male brains. In the striatum, cocaine upregulated NLRP3 inflammasome activity and CRF levels in males but not in females. In contrast, cocaine increased NLRP3 inflammasome activity in the HP of females but not in males and no effects on CRF signaling were observed in this region of either sex. Interestingly, cocaine increased NF-кB levels in the striatum and HP with no sex difference. Collectively, the authors provide evidence that cocaine can exert region- and sex- specific differences in neuroimmune signaling in the brain. Targeting neuroimmune signaling has been suggested as possible treatment for cocaine use disorders (CUDs). They suggest that sex should be taken into consideration when determining the efficacy of these new therapeutic approaches. 

The manuscript was well written and the conclusion was strongly supported by the experimental data. I have some minor considerations.

1. Abstract, Line 16, the "mice" should be replaced by "rats", because the authors conducted the experimental from the rats, but not mice in the present studies.

2. Methods, Lines 97 to 98, it is better to brief describe this model: how many days of cocaine self-administration? How amount of cocaine taken by each rat? When the brain of the rats were taken after self-administration? This information will guide the readers to understand the present results.

3. Results. It is better to use capitalized letters for the abbreviation of il, tnf, and ccl.

Author Response

In the present studies, the authors examined the effects of NOD-, LRR- and pyrin domain-containing protein 3 (NLRP3) inflammasome activity and corticotropin-releasing factor (CRF) signaling in self-administrative rats in response to cocaine. They found that cocaine activates microglia (Mg) in a region-specific manner in the brains of self-administered rats. To further characterize the effects of cocaine on Mg and neuroimmune signaling in vivo, they utilized the brains from both sexes of outbred rats with cocaine self-administration to explore the activation status of Mg, NLRP3 inflammasome activity, CRF signaling, and NF-кB levels in the striatum and hippocampus (HP). Saline self-administered rats of the same sex served as controls. They further found that cocaine increases neuroinflammation in the striatum and HP of both sexes with relatively higher fold increases in male brains. In the striatum, cocaine upregulated NLRP3 inflammasome activity and CRF levels in males but not in females. In contrast, cocaine increased NLRP3 inflammasome activity in the HP of females but not in males and no effects on CRF signaling were observed in this region of either sex. Interestingly, cocaine increased NF-кB levels in the striatum and HP with no sex difference. Collectively, the authors provide evidence that cocaine can exert region- and sex- specific differences in neuroimmune signaling in the brain. Targeting neuroimmune signaling has been suggested as possible treatment for cocaine use disorders (CUDs). They suggest that sex should be taken into consideration when determining the efficacy of these new therapeutic approaches.

The manuscript was well written and the conclusion was strongly supported by the experimental data. I have some minor considerations.

  1. Abstract, Line 16, the "mice" should be replaced by "rats", because the authors conducted the experimental from the rats, but not mice in the present studies.

Response: Sorry for the confusion.  That is our previous work which employed mice.  We made it clear in our abstract.      

 2. Methods, Lines 97 to 98, it is better to brief describe this model: how many days of cocaine self-administration? How amount of cocaine taken by each rat? When the brain of the rats were taken after self-administration? This information will guide the readers to understand the present results.

Response: Great point. Now the brief summary of cocaine self-administration regimen has been added in the method part.   

 3. Results. It is better to use capitalized letters for the abbreviation of il, tnf, and ccl.

Response: Yes. Change them.

Reviewer 3 Report

In this study, the authors explored possible regional- and sex-differences in cocaine-mediated neuroimmune dysregulation. Using RT-qPCR and Western blot, they provide evidence that cocaine can induce Microglia activation in the striatum and HP of both sexes. However, cocaine exerts region- and sex-specific effects on microglia activation, NLRP3 activity, and CRF signaling. Although the findings are interesting, more in-depth analysis is needed.

1. ASC speck or ASC oligomerization is the marker of NLRP3 inflammasome activation. Have the authors detected ASC speck in male and female rat brain before and after cocaine administration?

2    2.  Cocaine is capable of activating microglia in vitro and in vivo, the authors found microglia activation in male striatum but not in female striatum. The phenomenon is interesting, can the authors discuss the possible reason?

3   3.  Microglia or astrocytes play an important role in neuroinflammation signaling. Are there differences in the number of microglia or astrocytes in the brains of male and female rats before and after cocaine administration?

     4. In the discussion section, can the authors discuss the link between neuroinflammation and sex differences during CUD?

     5.  Are there differences in the behavior of male and female rats before and after cocaine administration?

Author Response

In this study, the authors explored possible regional- and sex-differences in cocaine-mediated neuroimmune dysregulation. Using RT-qPCR and Western blot, they provide evidence that cocaine can induce Microglia activation in the striatum and HP of both sexes. However, cocaine exerts region- and sex-specific effects on microglia activation, NLRP3 activity, and CRF signaling. Although the findings are interesting, more in-depth analysis is needed.

  1. ASC speck or ASC oligomerization is the marker of NLRP3 inflammasome activation. Have the authors detected ASC speck in male and female rat brain before and after cocaine administration?

Response: Great point. Now we have added ASC data by using the same samples into the context.

  1. Cocaine is capable of activating microglia in vitro and in vivo, the authors found microglia activation in male striatum but not in female striatum. The phenomenon is interesting, can the authors discuss the possible reason?

Response: The mechanisms underlying the phenomenon remain much elusive, still we have added some discussion in the main context. 

  1. Microglia or astrocytes play an important role in neuroinflammation signaling. Are there differences in the number of microglia or astrocytes in the brains of male and female rats before and after cocaine administration?

Response: We did not focus on the number changes of microglia and astrocytes in this project. We could do Iba1 or GFAP immunostaining in our future studies to answer this specific question.    

  1. In the discussion section, can the authors discuss the link between neuroinflammation and sex differences during CUD?

Response: Yes, now we have included more discussion in our main context. 

  1. Are there differences in the behavior of male and female rats before and after cocaine administration?

Response: Sorry we do not have the data for this question. 

Round 2

Reviewer 1 Report

Thanks to the authors for addressing some of the comments of the previous version. The manuscript is improved, but I still feel that this study is a correlative one. Also, if there is already a sex-related difference between male and female animals, any change after cocaine can be confounded with that, and strong statements should NOT be included.

Author Response

Thanks to the authors for addressing some of the comments of the previous version. The manuscript is improved, but I still feel that this study is a correlative one. Also, if there is already a sex-related difference between male and female animals, any change after cocaine can be confounded with that, and strong statements should NOT be included.

Responses: Thanks for the reviewer’s positive comments.  We agreed the reviewer’s comments and added some discussions.  We also tuned our voice down and deleted strong statements.